# Genetic Mapping and Molecular Characterization of a Broad-spectrum *Phytophthora sojae* Resistance Gene in Chinese Soybean

**DOI:** 10.3390/ijms20081809

**Published:** 2019-04-12

**Authors:** Chao Zhong, Yinping Li, Suli Sun, Canxing Duan, Zhendong Zhu

**Affiliations:** 1National Key Facility for Crop Gene Resources and Genetic Improvement, Institute of Crop Sciences, Chinese Academy of Agricultural Sciences, Beijing 100081, China; tonyzhong21@163.com (C.Z.); liyinpingxx@163.com (Y.L.); duancanxing@caas.cn (C.D.); 2Institute of Pomology, Chinese Academy of Agricultural Sciences, Xingcheng 125100, China

**Keywords:** next-generation sequencing, genetic mapping, *RpsX*, Phytophthora root rot, soybean

## Abstract

Phytophthora root rot (PRR) causes serious annual soybean yield losses worldwide. The most effective method to prevent PRR involves growing cultivars that possess genes conferring resistance to Phytophthora sojae (Rps). In this study, QTL-sequencing combined with genetic mapping was used to identify *RpsX* in soybean cultivar Xiu94-11 resistance to all *P. sojae* isolates tested, exhibiting broad-spectrum PRR resistance. Subsequent analysis revealed *RpsX* was located in the 242-kb genomic region spanning the *RpsQ* locus. However, a phylogenetic investigation indicated Xiu94-11 carrying *RpsX* is distantly related to the cultivars containing *RpsQ*, implying *RpsX* and *RpsQ* have different origins. An examination of candidate genes revealed *RpsX* and *RpsQ* share common nonsynonymous SNP and a 144-bp insertion in the *Glyma.03g027200* sequence encoding a leucine-rich repeat (LRR) region. *Glyma.03g027200* was considered to be the likely candidate gene of *RpsQ* and *RpsX*. Sequence analyses confirmed that the 144-bp insertion caused by an unequal exchange resulted in two additional LRR-encoding fragments in the candidate gene. A marker developed based on the 144-bp insertion was used to analyze the genetic population and germplasm, and proved to be useful for identifying the *RpsX* and *RpsQ* alleles. This study implies that the number of LRR units in the LRR domain may be important for PRR resistance in soybean.

## 1. Introduction

Soybean [*Glycine max* (L.) Merr] is one of the most important economic crops, accounting for more than half of the global oilseed production [1]. Sustainable soybean cultivation is largely limited by diseases caused by diverse pathogens, including the soil-borne oomycete *Phytophthora sojae*, which causes Phytophthora root rot (PRR), with annual economic losses of 1–2 billion worldwide [2,3,4,5,6]. In China, PRR was first detected in Heilongjiang province in 1989, and it has since spread to most soybean-producing areas [7]. This disease can occur at any soybean plant developmental stage. In water-saturated soils, *P. sojae* zoospores can infect soybean plants throughout the growing season, causing damping-off before and after emergence, root and stem decay during the adult stage, and even death [8]. If PRR is established in the field, it is difficult to control with chemical and biological agents, ultimately resulting in considerable or complete yield losses [7]. The most effective way to control PRR currently involves growing soybean cultivars expressing *Rps* genes, which confer resistance to *P. sojae* [9]. Compared with the application of biological and chemical agents, growing PRR-resistant cultivars is a more environmentally friendly method for the sustainable development and production of crops.

The two types of PRR resistance have been identified in soybean, including partial resistance which is controlled by multiple genes, and complete resistance which is mediated by the single dominant *Rps* resistance gene [9]. Soybean plants exhibiting partial resistance restrict the spread of *P. sojae* in plant tissues, whereas plants displaying complete resistance are immune to *P. sojae*. Furthermore, the single dominant *Rps* genes mediating complete resistance can easily be identified and incorporated into other soybean cultivars for PRR control [9,10,11,12]. Thirty-two *Rps* genes have been identified to date, and they have been localized to nine soybean chromosomes [9,13,14,15,16,17,18,19,20,21,22,23,24,25]. However, *Rps* genes are race-specific and conform to the gene-for-gene hypothesis with a *P. sojae* avirulence gene. Several studies have revealed that the virulence of a *P. sojae* population rapidly changes, and a single race can generate various pathotypes that can overcome the resistance mediated by the *Rps* genes [9,26,27,28,29,30,31,32]. Due to the breakdown of resistance caused by the emergence of new *P. sojae* pathotypes, the durability of an *Rps* gene is generally only 8–15 years [9,28]. Therefore, researchers must continuously search for and identify new *Rps* genes.

The *Rps* genes have traditionally been identified and mapped to construct the corresponding mapping populations, including the F_2:3_, recombinant inbred line, and near-isogenic line populations. Additionally, the mapping interval containing the *Rps* genes was determined by analyzing the phenotypes and polymorphic molecular markers in the segregating mapping population [18,19,20,21]. As this process necessitates the screening of many polymorphic markers between the parents and the extreme-phenotype bulks as well as the genotyping of a relatively large population, it is labor intensive and requires considerable time and resources. Recent advances in next-generation sequencing (NGS) technology have gradually decreased the costs associated with sequencing and increased the application of NGS-based methods for genomics studies related to crop improvement [33,34]. The sequencing, assembly, and annotation of the Williams82 soybean reference genome have enabled the identification of increasing numbers of single nucleotide polymorphisms (SNPs) and insertions/deletions (InDels) through whole-genome resequencing and facilitated the development of key markers and the detection of *Rps* candidate genes [18,19,20,24,35,36,37]. Moreover, quantitative trait loci sequencing (QTL-seq) is a relatively new method that combines NGS technology with bulk segregant analysis, which is a rapid and effective approach for identifying markers linked to specific traits, to rapidly detect candidate genomic locations [38]. Identifying candidate genomic regions with sequencing data for the extreme-phenotype bulks of 20–50 selected lines has become a popular approach. The QTL-seq method has been used to map qualitative and quantitative traits in multiple plant species [39,40,41,42,43,44,45]. We recently identified and mapped a novel *Rps* gene, *RpsHC18*, on chromosome 3 through a combination of QTL-seq and traditional genetic mapping [24]. Thus, this is an important new method for efficiently identifying and mapping *Rps* genes.

China has abundant germplasm resources resistant to PRR, and some *Rps* genes in the corresponding cultivars/landraces have been previously reported [13,14,16,20,21,24]. Soybean cultivar Xiu94-11, which is rich in oil, was developed in Liaoning province in China. While identifying and screening PRR-resistant resources, we determined that Xiu94-11 is highly resistant to *P. sojae* isolates in China. Hence, the objectives of this study were to (1) characterize the inheritance of PRR resistance in Xiu94-11, (2) map the *Rps* gene in Xiu94-11 through QTL-seq and genetic mapping, and (3) identify the *Rps* candidate genes and their functional markers.

## 2. Results

### 2.1. Xiu94-11 Has Broad-spectrum Resistance to *Phytophthora sojae* Which is Controlled by a Dominant Single Gene

The resistance to 14 *P. sojae* isolates with varying virulence levels was assessed for the PRR-resistant Xiu94-11 and 22 other cultivars each containing a different identified *Rps* gene as well as four PRR-susceptible cultivars (Zhonghuang13, Williams, Zhonghuang47, and Jikedou2) as controls. All of the soybean plants of the PRR-susceptible cultivars died after the inoculations with the *P. sojae* isolates. Moreover, 13 reaction types were observed among the 27 tested soybean cultivars in response to the 14 *P. sojae* isolates (Table 1). Specifically, Xiu94-11 was resistant to all 14 *P. sojae* isolates and exhibited the broadest spectrum resistance among the PRR-resistant cultivars. Thus, Xiu94-11 may contain a novel *Rps* gene or a unique combination of the identified *Rps* genes.

To analyze the genetic characteristics underlying the resistance of Xiu94-11 and further map the *Rps* gene(s) involved, an F_2:3_ mapping population was derived from a hybridization between Zhonghuang47 and Xiu94-11. *Phytophthora sojae* isolates PsMC1 and PsJS2 were used to evaluate the phenotypes of 137 F_2:3_ families and parental cultivars. All families exhibited consistent responses to the two *P. sojae* isolates. The 38 homozygous resistant families, 63 segregating families, and 36 susceptible families fit the expected 1:2:1 ratio (Table 2). These results suggested that the resistance of Xiu94-11 is controlled by a single dominant gene, which we tentatively named *RpsX*.

### 2.2. The Resistance Gene *RpsX* in Xiu94-11 is Located on Soybean Chromosome 3

Genomic DNA samples of Xiu94-11, Zhonghuang47, and PRR-resistant and -susceptible bulks underwent whole-genome resequencing based on an Illumina system. The PRR-resistant parent, Xiu94-11, generated 24.4 Gb clean reads, with an average depth of 22× and 99.33% 5× genome coverage, whereas the susceptible parent, Zhonghuang47, produced 24.3 Gb clean reads, with an average depth of 21× and 99.16% 5× genome coverage (Appendix A). The PRR-resistant bulk (R30) consisting of an equal amount of DNA from 30 homozygous resistant families produced 61.5 Gb clean reads, with an average depth of 61× and 99.10% 5× genome coverage. The PRR-susceptible bulk (S30) representing 30 homozygous susceptible families produced 65.2 Gb clean reads, with an average depth of 65× and 99.20% 5× genome coverage (Appendix A).

The QTL-seq approach was used to identify the genomic region containing *RpsX* [38]. A total of 1,159,236 high-quality SNPs were obtained, and the delta SNP index of each SNP was calculated. The distribution of the delta SNP index (R30–S30) on 20 chromosomes (Appendix A) at the 99% confidence level revealed a contiguous region exceeding the threshold in the 1.05–3.55 Mb genomic region of chromosome 3 (Figure 1). Accordingly, this region represented the only candidate region for *RpsX*.

### 2.3. *RpsX* Was Finely Mapped to the 242 kb Region on Chromosome 3

To validate the accuracy of the *RpsX* candidate region identified by QTL-seq and further limit the *RpsX* genomic interval, a genetic mapping approach was used to analyze all 137 F_2:3_ families. The published simple sequence repeat (SSR) markers in the candidate region and the InDel markers developed based on the InDels identified via whole-genome resequencing were used for screening the polymorphism between the parental cultivars and clarifying the genotypes of the populations [24,25,37]. Among 150 SSR markers, seven identified polymorphisms between Xiu94-11 and Zhonghuang47 and were closely linked to *RpsX* (Figure 2A). Additionally, 20 InDels between Xiu94-11 and Zhonghuang47 were developed as PCR-based markers, four of which were associated with polymorphisms and were tightly linked to *RpsX* (Figure 2A). Moreover, *RpsX* was mapped between the InDel marker InDelxz6 and the SSR marker BARCSOYSSR_03_0175, with genetic distances of 0.4 and 0.7 cM, respectively, and co-segregated with three SSR markers (BARCSOYSSR_03_0161, BARCSOYSSR_03_0165, and BARCSOYSSR_03_0167). On the basis of the physical position of each marker on chromosome 3, the genomic region comprising *RpsX* was localized to a 242-kb region (2,910,913–3,153,254 bp) (Figure 2B), which spans the *RpsQ* genomic region (Figure 2C) [20]. Polymorphisms were screened and genetic linkages were analyzed with the published InDel markers (Insert144 and Insert11) that reportedly co-segregate with *RpsQ* [20]. The resulting data indicated that the two markers also co-segregated with *RpX*. Therefore, we speculated that *RpsX* may be an allele of *RpsQ*.

### 2.4. RpsX and RpsQShared the Same Candidate Gene Model

The available information regarding the annotated soybean genome (*Glyma.Wm82.a2.v1*) indicates there are 24 gene models in the *RpsX* mapping interval (https://www.soybase.org/). Fourteen nonsynonymous SNPs (nsSNPs) with a delta SNP index of 1 were identified distributed in eight gene models between the resistant and susceptible bulks (Figure 3). Among the eight gene models containing nsSNPs, *Glyma.03g027200* containing five nsSNPs and annotated as a serine/threonine protein kinase (STK) with leucine-rich repeats (LRRs) is reportedly a type of plant resistance gene. In contrast, there are no reports suggesting the other seven gene models are related to disease resistance in plants. Therefore, *Glyma.03g027200* was identified as the most likely *RpsX* candidate gene. Among the analyzed cultivars, the *RpsX* mapping interval of Xiu94-11 (*RpsX*) had the same nsSNP as the corresponding intervals of Qichadou1 (*RpsQ*) and the parental cultivar of Qichadou1, Ludou4 (*Rps9*), only for *Glyma.03g027200* (Figure 3).

As *Glyma.03g027200* was identified as an *RpsQ* candidate gene, we examined whether Xiu94-11 and Qichadou1 are related and whether *RpsX* and *RpsQ* are actually the same gene. We completed a phylogenetic analysis based on the SNPs identified by NGS among 45 cultivars/landraces, some of which are related to Xiu94-11 and Qichadou1 (Appendix A). A total of 48,049 SNPs on chromosome 3 identified among 45 cultivars were selected to construct a phylogenetic tree according to the neighbor-joining method. We observed that 45 soybean genotypes formed two subgroups, and Xiu94-11 belonged to a subgroup separate from that of Ludou4 and Qichadou1 (Figure 4). The cultivar most closely related to Xiu94-11 was Kaohsiung1, which was selected from Chu-tzu-dow, a landrace from Taiwan, which is not related to Qichadou1 [46]. These results implied that Xiu94-11 is not closely related to Qichadou1.

### 2.5. 144-bp Insertion in LRR Domain is Present in Resistant Haplotype of RpsX Locus

To further explore the candidate gene sequence and structural differences between the resistant and susceptible genotypes, the allele sequences of *Glyma.03g027200* in Xiu94-11 and Zhonghuang47 were obtained with a Sanger sequencing method. All five nsSNPs identified by NGS were confirmed in Xiu94-11 and Zhonghuang47. A comparison of the obtained and the previously published *RpsQ* candidate gene sequences [20] revealed a 99% sequence identity between the *RpsX* and *RpsQ* alleles. Additionally, we detected a 97% sequence identity between the *RpsX* allele and the alleles of the Williams82 reference genome sequence and the susceptible control Zhonghuang41 (Appendix A). These results indicated that the *Glyma.03g027200* sequence is relatively conserved and is highly similar between the resistant and susceptible genotypes. The deduced protein sequences based on the genomic sequences were aligned and analyzed for conserved domains. We detected two functional domains in all aligned sequences, namely the LRR and STK domains. However, the resistant and susceptible alleles differ in the sequence encoding the LRR region. Specifically, 11 LRR motifs are encoded in the *RpsX* and *RpsQ* alleles of the resistant genotype, whereas nine LRR motifs are encoded in the corresponding alleles of the susceptible genotypes. Like the *RpsQ* allele, the *RpsX* allele contains a 144-bp insertion, resulting in the insertion of 48 amino acid residues comprising two LRR structural units (Figure 5). The genomic sequence most similar to the 144-bp insertion is a 144-bp sequence upstream of the candidate gene, implying the insertion may have been the result of a replication of the upstream fragment (Figure 5).

Because there are currently no soybean cultivars derived from Xiu94-11, to further verify whether the detected 144-bp insertion is important for the resistance to *P. sojae*, soybean genotypes related to Qichadou1 based on pedigrees were analyzed regarding their reactions to 12 *P. sojae* isolates. Additionally, their alleles corresponding to *Glyma.03g027200* were sequenced (Appendix A). A phylogenetic tree was constructed according to the neighbor-joining method using the *Glyma.03g027200* allelic sequences of 30 genotypes. Four cultivars, namely Xiu94-11 (*RpsX*), Qichadou1 (*RpsX*), Ludou4 (*Rps9*), and Kexin5, were clustered in one subgroup. Moreover, the same 144-bp insertion was detected in their *Glyma.03g027200* allelic sequences (Figure 6). Qichadou1 was derived from a cross between Ludou4 and Peking, whereas Kexin5 was the result of the chemical mutagenesis of Ludou4. These four cultivars all exhibited excellent resistance to 10–12 *P. sojae* isolates, suggesting that the 144-bp insertion in the sequence encoding the LRR region may be important for the observed resistance to *P. sojae*.

### 2.6. Developed Marker Insert144 Is Able to Efficiently Detect Resistant Haplotypes at RpsX Locus

Because the 144-bp insertion is a key variant of the *RpsX* locus, Insert144, which is an InDel marker based on this insertion, was developed as a marker that co-segregates with *RpsQ* and was used to distinguish *RpsQ* from the *Rps1* alleles. In the present study, Insert144 was further validated in all cultivars carrying currently identified *Rps* genes as well as in the susceptible controls [20]. Only three soybean cultivars, namely Xiu94-11 (*RpsX*), Ludou4, and Qichadou1 (*RpsQ*), contained this insertion (Figure 7). The screening and identification of *RpsX* and its alleles among 177 soybean germplasms revealed that six soybean cultivars (Ludou2, Qihuang9, Fendou78, Fendou79, Qihuang12, and Qihuang13) had the same genotype as Xiu94-11. The phenotypic responses to eight *P. sojae* isolates indicated that they were all resistant to PRR, meaning the eight cultivars are likely to contain *RpsX* or PRR-resistance alleles at the *RpsX* locus. These results suggested that Insert144 can serve as a functional and diagnostic marker for *RpsX* and the PRR-resistance alleles during soybean breeding.

## 3. Discussion

In this study, a rapid approach combining high-throughput sequencing and traditional genetic mapping was deployed to identify a novel allele, *RpsX*, at the *RpsQ* locus in a small F_2:3_ population. The most prominent advantage of this approach is that it enables the genotyping and mapping of a resistance gene in a relatively early generation like F_2:3_. Additionally, the SNPs and InDels identified based on high-throughput sequencing can be used for further fine mapping, a haplotype analysis, and the identification of candidate genes [24,25,38,39,40,41,42,43,44,45]. *Phytophthora sojae* virulence involves a complex mechanism and can rapidly change, enabling this pathogen to quickly overcome the resistance conferred by most *Rps* genes, ultimately leading to severe yield losses. Thus, whole-genome resequencing represents a fast, efficient, accurate, and relatively simple method for identifying novel *Rps* genes [24,25].

Many of the currently known *Rps* genes have been mapped on the short arm of soybean chromosome 3, including some genes that have been finely mapped such as *Rps1k*, *RpsYD29*, *RpsQ*, *RpsHC18*, *RpsWY*, *RpsHN*, and *RpsUN1* [11,20,24,25,26,27,30]. Moreover, most of these genes were mapped in the interval containing typical plant resistance genes encoding a nucleotide-binding site (NBS) and an LRR domain. Previous studies indicated that some genes encoding an NBS-LRR structure were candidate genes for these *Rps* genes [14,18,21,24]. The *Rps1k* gene, which is associated with broad-spectrum and durable resistance, has been cloned, and the tandemly arranged NBS-LRR genes *Rps1k-1* and *Rps1k-2* were functionally validated as responsible for conferring complete resistance to *P. sojae* [47,48]. Unlike the *Rps* genes in tandemly arranged NBS-LRR gene clusters, *RpsX* was mapped to an interval on the short arm of chromosome 3 lacking NBS-LRR genes. Only one plant resistance gene, *Glyma.03g027200*, which encodes an STK-LRR structure, was detected in the *RpsX* region. Additionally, *Glyma.03g027200* contains an *RpsX*-specific nsSNP, which was identified by QTL-seq (Figure 3). These results imply that *Glyma.03g027200* is the likely candidate gene for *RpsX,* and may represent another gene type conferring resistance to *P. sojae*.

Interestingly, *Glyma.03g027200* is also the candidate gene for *RpsQ*, which we previously identified and mapped [20]. The majority of the mapping intervals with *RpsX* coincided with the mapping intervals with *RpsQ* (Figure 2B,C). A pedigree analysis revealed that Qichadou1, which contains *RpsQ*, was the result of a hybridization between Ludou4 and Peking, which is a PRR-susceptible cultivar [20]. In contrast, Xiu94-11 was obtained from a cross between lines 89-6 and Dandou806. Both of these lines were derived from landraces originating in northeastern China, whereas the ancestors of Qichadou1 were landraces originating in Shandong province, China, a Peking landrace, and the American cultivar Magnolia. Consequently, Xiu94-11 and Qichadou1 are not related. Moreover, our phylogenetic analysis based on the homozygous SNPs on chromosome 3 revealed a distant genetic relationship between Xiu94-11 and Qichadou1 and Ludou4. Interestingly, however, the *Glyma.03g027200* allele sequences of *RpsX* and *RpsQ* are highly similar and carry the same SNPs and InDels. Therefore, we speculated that *RpsX* and *RpsQ* evolved independently via the same molecular mechanism in different ecological regions. The *Glyma.03g027200* locus should be a mutation hotspot in the soybean genome, and additional PRR-resistance alleles may exist in soybean germplasm.

The *Glyma.03g027200* sequence comprises an STK-LRR gene, which is another type of plant resistance gene. The STK-LRR resistance gene encodes an extracellular receptor-like protein kinase with an extracellular LRR and an intracellular STK. A few plant resistance genes have been cloned and verified to be STK-LRR genes, including rice genes conferring resistance to bacterial blight (*Xa21*, *Xa21D*, and *Xa3*/*Xa26*), a wheat leaf rust resistance gene (*Lr10*), and an apple scab resistance candidate gene (*Rvi12_Cd5*) [49,50,51,52,53]. In the present study, the candidate gene sequences were highly similar (97%) between the PRR-resistant genotypes Xiu94-11 (*RpsX*) and Qichadou1 (*RpsQ*) and the PRR-susceptible genotypes Zhonghuang47 and Williams82. Most of the sequence differences between the resistant and susceptible genotypes were detected in the sequence encoding the LRR domain, which plays a key role in the disease resistance mechanism of plants. The LRR domain determines the specificity of the recognition of pathogen effectors [54,55]. The LRR units containing approximately 20–30 amino acid residues consist of repeating core xxLxLxx motifs (L = leucine or other aliphatic amino acids, x = any amino acid) [56]. Allelic mutations due to SNPs and InDels in the sequence encoding the LRR domain may result in the generation of new R genes [57,58]. Because of the high sequence identity between the sequences encoding the LRR units in the LRR domain, unequal crossing-over and illegitimate recombinations are prone to occur, resulting in new R gene specificities due to the differences in the number of encoded LRR units [56,59,60]. Therefore, in this study, the 144-bp insertion of the *RpsX* and *RpsQ* candidate genes was caused by an unequal exchange with the adjacent 144-bp fragment, resulting in a new *Rps* gene.

InDel marker Insert144 is a co-segregated marker developed to detect 144-bp insertions in *RpsQ*. Additionally, this marker can distinguish *RpsQ* from the alleles at the *Rps1* locus. Thus, this marker co-segregates with and can be used to detect *RpsX*. We used this marker to analyze diverse soybean genotypes and determined that it can detect PRR-resistance alleles at the *RpsX/Q* locus. Therefore, Insert144 may be applied for molecular marker-assisted selection. The PRR-resistant genotypes containing the 144-bp insertion are all highly resistant to *P. sojae*, indicating this insertion is important for the resistance to *P. sojae*.

In conclusion, we identified a novel allele, *RpsX*, at the *RpsQ* locus using a QTL-seq method involving high-throughput sequencing and traditional genetic mapping. A cluster analysis with homozygous SNPs on chromosome 3 and an analysis of the allelic sequences of the candidate gene confirmed that the genetic background of *RpsX* and *RpsQ* varies considerably, but both candidate alleles have the same 144-bp insertion in the sequence encoding the LRR domain. Therefore, changes to the LRR-encoding region play an important role in the development of novel *Rps* genes. The method described herein represents a rapid and efficient procedure for identifying novel *Rps* genes and may be useful for the cloning of *RpsX* and *RpsQ* as well as the application of *RpsX* and *RpsQ* functional markers for marker-assisted selection.

## 4. Materials and Methods

### 4.1. Phenotyping for PRR Resistance

Phytophthora root rot-resistant cultivar Xiu94-11 along with 22 cultivars with a single *Rps* gene and four PRR-susceptible cultivars were analyzed regarding their resistance to 14 *P. sojae* isolates that varied in terms of virulence. Additionally, 28 soybean cultivars and landraces related to Qichadou1 as well as Xiu94-11 and Zhonghuang47 were inoculated with 12 *P. sojae* isolates (Figure 6) to identify the *RpsX* PRR-resistance alleles. Soybean plants were grown and inoculated with *P. sojae* isolates as previously described [24,25].

A mapping population was constructed with Zhonghuang47 as the female parent and Xiu94-11 as the male parent to derive F_1_ seeds. An F_1_ seed was used to produce 137 F_2:3_ families by self-crossing. For each family, 20–25 seeds were sown in paper cups filled with vermiculite to evaluate the responses of the resulting plants to *P. sojae* isolates PsMC1 and PsJS2. The plants were inoculated and their phenotypes were evaluated as previously described [24,25].

### 4.2. Next-generation Sequencing and QTL-seq Analysis of Resistant and Susceptible Bulks

On the basis of the phenotypic evaluations, 30 homozygous resistant and 30 susceptible families were respectively used to construct PRR-resistant (R30) and PRR-susceptible (S30) bulks for a subsequent NGS with an Illumina system. The generated data underwent a QTL-seq analysis [25,38]. The DNA of the extreme-phenotype bulks and the Xiu94-11 and Zhonghuang47 cultivars were isolated with the Plant Genomic DNA Kit (Tiangen, Beijing, China). The raw read data generated for the Illumina libraries were filtered to produce clean reads, which were then aligned with the *Glyma.Wm82.a2.v1* reference genome (http://phytozome.jgi.doe.gov/pz/portal.html) with the genome alignment software BWA [61]. The SNPs and InDels in the two extreme-phenotype bulks and parental cultivars were detected and filtered with the variation analysis software GATK [24,61].

The SNP index of each bulk and the delta SNP index were calculated based on an SNP detected in the two extreme-phenotype bulks during a previous filtering analysis [38]. Three confidence levels (*P* < 0.1, 0.05, and 0.01) were set for the delta SNP index [38], which was calculated for each SNP position with the following formula:
delta SNP index = SNP index (PRR-resistant bulk) − SNP index (PRR-susceptible bulk).

### 4.3. Linkage Analysis and Genetic Mapping of the Candidate Region

After identifying the *RpsX* candidate region via QTL-seq, a genetic linkage mapping approach was used to further limit the *RpsX* candidate region. Publicly available SSR markers in the *RpsX* candidate region were selected to screen for polymorphisms between the parental cultivars and the genotype mapping population [47]. High-quality InDels identified by whole-genome resequencing were further developed to PCR markers to map the candidate region (Appendix A) [24,25]. Combining phenotypic and genotypic results, the genetic linkage analysis of *RpsX* was completed with the MAPMAKER/EXP (version 3.0) program [62]. The genetic linkage map was constructed with the MapDraw program [63].

### 4.4. Phylogenetic Analysis of Soybean Genotypes

Forty-five soybean genotypes, including landraces/cultivars, were selected to analyze the genetic relationship between Xiu94-11 and Qichadou1/Ludou4. Moreover, in addition to Xiu94-11 and Zhonghuang47, 43 soybean genotypes were subjected to whole-genome resequencing (unpublished data). The DNA of these genotypes was extracted and sent to Annoroad Gene Technology (Beijing, China) for the construction of Illumina sequencing libraries. Raw data were filtered, sequences were aligned with reference genome sequences, and homozygous SNPs were identified and annotated as previously described [24,25]. The homozygous SNPs identified on chromosome 3 for each genotype were used to construct a phylogenetic tree according to the neighbor-joining method of the MEGA 6.0 program [64].

### 4.5. Analysis of the Allelic Sequences of the Candidate Gene Locus

The allele sequences of the *Glyma.03g027200* locus for the cultivars related to Qichadou 1 were determined by PCR-based Sanger sequencing. The sequencing primers were the same primer pairs used to amplify the *RpsQ* candidate allele and overlapped new primers designed with the NCBI Primer-BLAST tool (https://www.ncbi.nlm.nih.gov/tools/primer-blast/) (Appendix A). A PCR assay was completed with PrimeSTAR™ HS DNA Polymerase (Takara Biotechnology, Dalian, China). The PCR product obtained for each sample was sequenced by Sangon Biotech (Beijing, China). The resulting sequences were assembled with ContigExpress to obtain the complete *Glyma.03g027200* allele sequences. Each sample was amplified and sequenced three times to avoid errors generated during amplification and sequencing.

A multiple sequence alignment involving the obtained sequences was completed with ClustalW, after which a phylogenetic tree was constructed according to the neighbor-joining method (with 1000 bootstrap replicates) of the MEGA 6.0 program [64]. The sequence coding region was confirmed based on the cDNA sequence of the *RpsQ* allele and the published annotated Williams82 reference transcripts (https://www.soybase.org/). The coding region sequence was converted to a protein sequence with ExPASy (https://web.expasy.org/translate/). Conserved domains were predicted for the Xiu94-11, Zhonghuang47, Qichadou1, Ludou4, and Williams82 sequences with the Conserved Domain Database (http://www.ncbi.nlm.nih.gov/cdd/) [65] and SMART (http://smart.embl-heidelberg.de/) [66].

### 4.6. Validation and Screening of *RpsX* and Its Alleles in Soybean Genotypes Using the Functional Marker Insert144

Cultivars containing single *Rps* genes were genotyped with the Insert144 marker that co-segregated with *RpsQ*. This marker was developed based on the 144-bp insertion in the *RpsQ* allele. Insert144 was also used to detect the *RpsX* and *RpsQ/9* resistance alleles in 177 soybean cultivars and landraces whose reactions to eight *P. sojae* isolates had been analyzed. The PCR assay was completed as previously described.

## Figures and Tables

**Figure 1 ijms-20-01809-f001:**
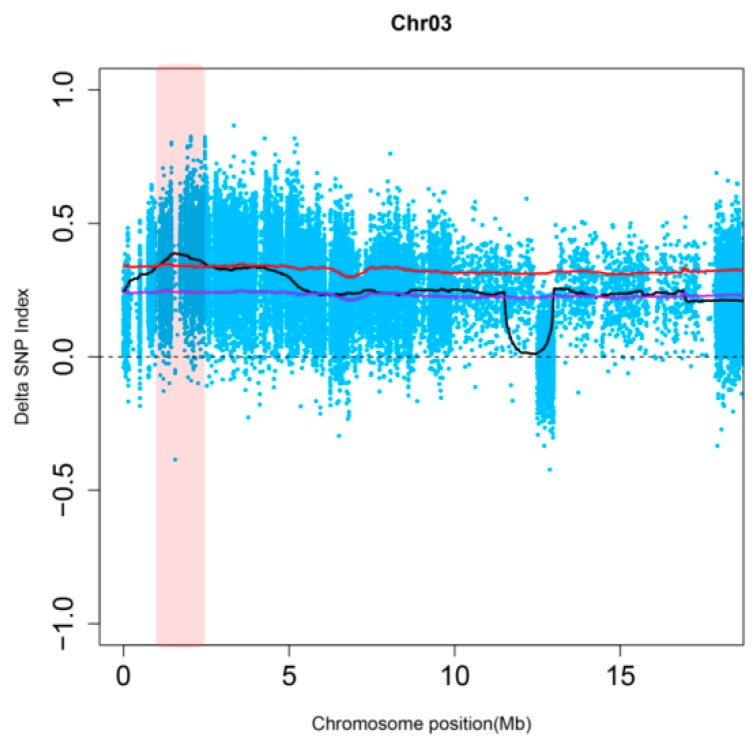
Delta Single nucleotide polymorphisms (SNP) index of chromosome 3. The blue dots represent the delta SNP index corresponding to an SNP obtained by filtering the two bulks. The black, purple, and red lines respectively represent the average value of the delta SNP index as well as the 95% and 99% confidence level thresholds in the corresponding window calculated by the sliding window method. The distribution of the delta SNP index at the 99% confidence level revealed only one contiguous region exceeding the threshold in the 1.05–3.55 Mb genomic region (red) of chromosome 3. The window is 1 Mb, with 10-kb slides.

**Figure 2 ijms-20-01809-f002:**
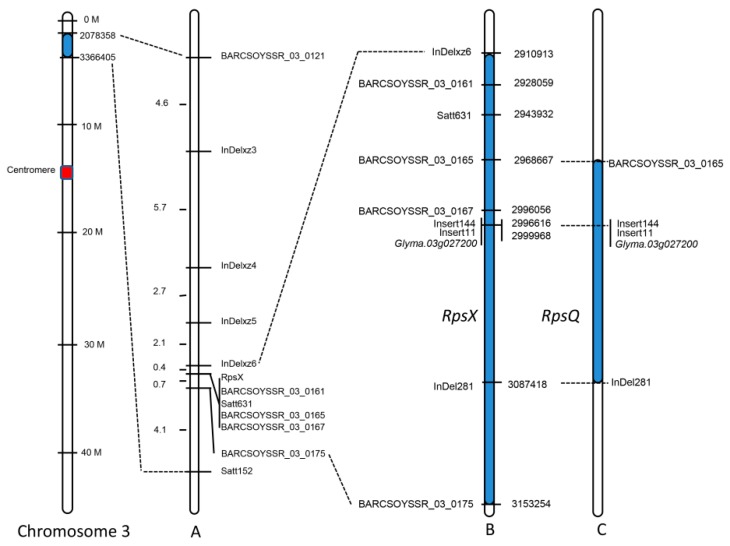
Genetic and physical maps of the *RpsX* region. (**A**) Genetic linkage analysis of *RpsX* based on the F_2:3_ mapping population. The genetic distances (cM) are provided on the left, and marker locations are indicated on the right. (**B**) Candidate physical interval of *RpsX* (blue region) and the physical positions of the linked and co-segregated markers of *RpsX*. (**C**) *RpsQ* mapping interval and the corresponding markers [20].

**Figure 3 ijms-20-01809-f003:**
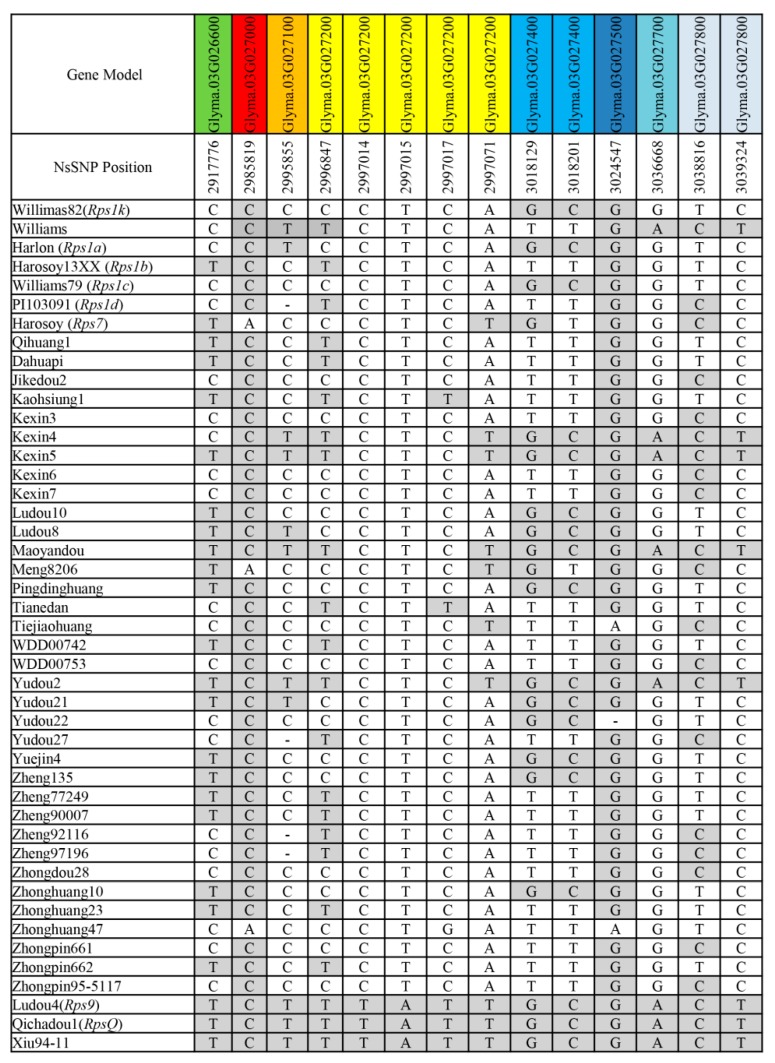
Nonsynonymous SNPs located in the corresponding gene models of the *RpsX* mapping interval among 45 soybean genotypes. Different gene models are presented in different colors. A base in a white background is the same as that in the PRR-susceptible parent Zhonghuang47, whereas a base in a gray background is the same as that in the PRR-resistant parent Xiu94-11.

**Figure 4 ijms-20-01809-f004:**
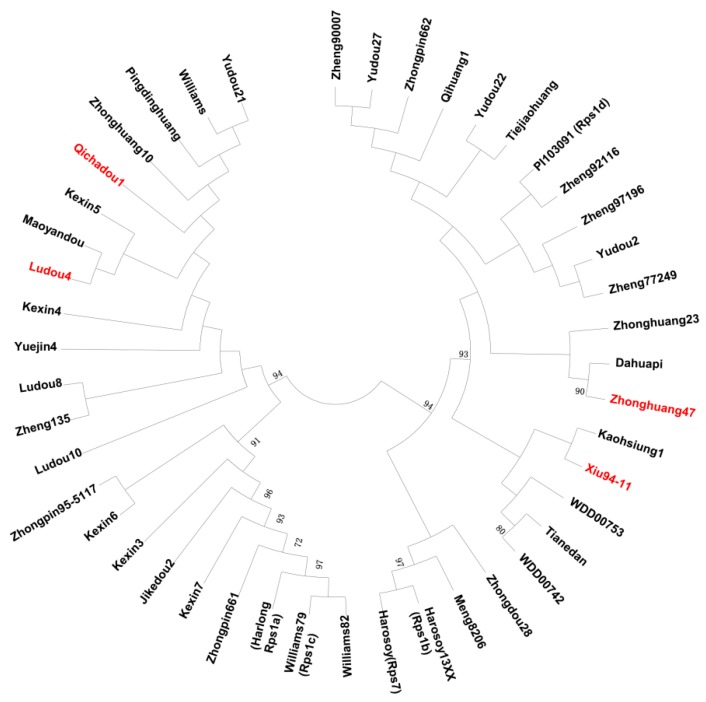
Phylogenetic tree constructed according to the neighbor-joining method using homozygous SNPs identified on chromosome 3 among 45 soybean genotypes, including cultivars and landraces.

**Figure 5 ijms-20-01809-f005:**
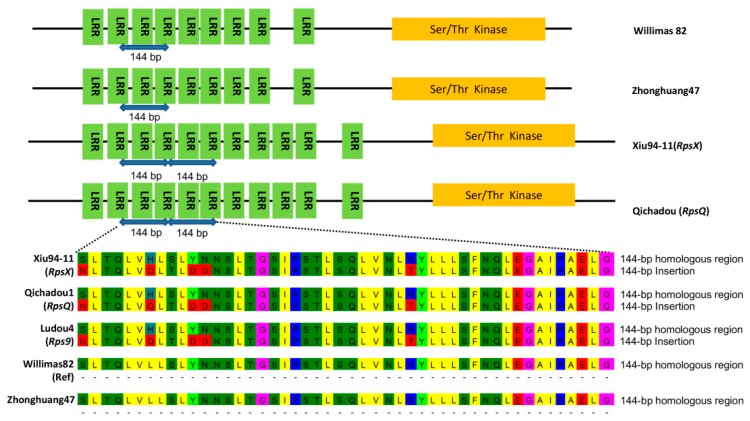
Analysis of conserved domains encoded by the alleles of soybean genotypes. The *RpsX* and *RpsQ* alleles contain a 144-bp insertion encoding 48 amino acid residues. This insertion is most similar to a 144-bp sequence upstream of the candidate gene.

**Figure 6 ijms-20-01809-f006:**
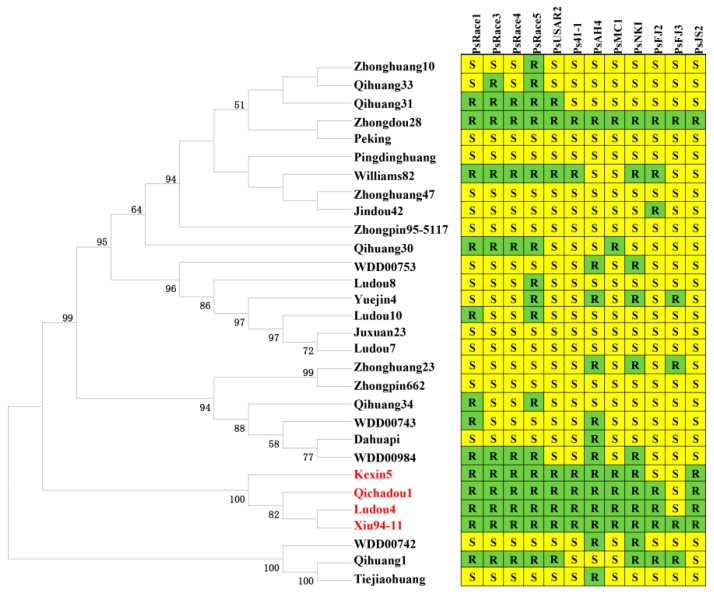
Phylogenetic tree constructed according to the neighbor-joining method using the allelic sequences of the candidate gene model *Glyma.03g027200* in 30 soybean cultivars. The reactions of the cultivars to 12 *P. sojae* isolates are also presented.

**Figure 7 ijms-20-01809-f007:**
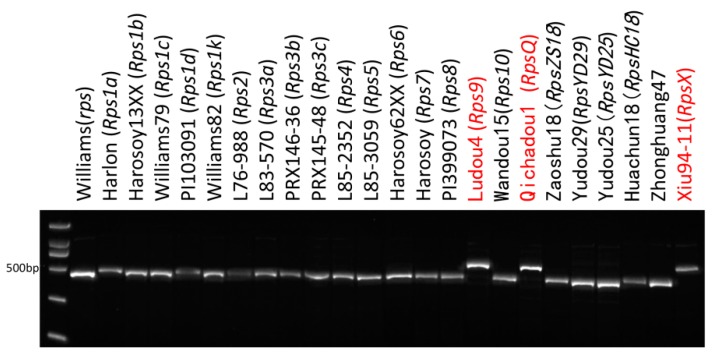
Molecular banding patterns for PRR-resistant cultivars containing the identified *Rps* genes. The cultivars were analyzed with the Insert144 marker developed based on the 144-bp insertion in *RpsX* and *RpsQ*.

**Table 1 ijms-20-01809-t001:** Phenotypic responses of 27 soybean cultivars to 14 *Phytophthora sojae* isolates.

Cultivar (*Rps* gene)	PsRace1	PsRace3	PsRace4	PsRace5	PsUSAR2	Ps41-1	PsAH4	PsMC1	PsNKI	PsFJ2	PsFJ3	PsJS2	Ps6497	Ps7063
Harlon (*Rps1a*)	S ^1^	S	S	S	R	S	R	S	R	S	S	S	R	S
Harosoy13XX (*Rps1b*)	R	R	R	R	S	S	S	S	S	S	S	S	S	R
Williams79 (*Rps1c*)	R	R	R	R	R	R	S	R	R	R	R	S	R	R
PI103091 (*Rps1d*)	R	S	S	R	R	S	S	S	S	S	S	S	S	S
Williams82 (*Rps1k*)	R	R	R	R	R	R	S	S	R	R	S	S	R	R
L76-988 (*Rps2*)	R	R	R	R	S	S	S	S	S	S	S	S	S	S
L83-570 (*Rps3a*)	R	R	R	R	R	S	S	S	S	S	S	S	R	S
PRX146-36 (*Rps3b*)	R	R	S	R	R	S	S	S	S	S	S	S	R	R
PRX145-48 (*Rps3c*)	R	R	R	R	S	S	S	S	S	S	S	S	S	R
L85-2352 (*Rps4*)	R	R	R	R	R	S	S	S	S	S	S	S	R	S
L85-3059 (*Rps5*)	R	R	R	R	S	S	S	S	S	S	S	S	R	S
Harosoy62XX (*Rps6*)	R	R	R	R	R	S	R	S	S	S	S	S	R	S
Harosoy (*Rps7*)	R	R	R	S	R	S	S	S	S	S	S	S	S	S
PI399073 (*Rps8*)	R	R	R	R	R	S	R	S	S	S	S	S	R	S
Youbian30 (*RpsYB30*)	R	R	S	R	R	R	S	S	R	R	S	S	S	S
Yudou25 (*RpsYD25*)	R	R	R	R	R	R	S	R	R	S	R	S	R	R
Yudou29 (*RpsYD29*)	R	R	R	R	R	R	S	R	R	R	R	S	R	R
Ludou4 (*Rps9*)	R	R	R	R	R	R	R	R	R	R	S	R	R	R
Qichadou 1 (*RpsQ*)	R	R	R	R	R	R	R	R	R	R	S	R	R	R
Wandou15 (*Rps10*)	R	R	R	R	R	R	R	R	R	S	R	S	R	S
Zaoshu18 (*RpsZS18*)	R	R	R	R	R	R	S	S	R	R	R	S	R	S
Huachun18 (*RpsHC18*)	R	R	R	R	R	R	R	R	R	R	R	R	R	S
Xiu94-11 (*RpsX*)	R	R	R	R	R	R	R	R	R	R	R	R	R	R
Zhonghuang13	S	S	S	S	S	S	S	S	S	S	S	S	S	S
Williams (*rps*)	S	S	S	S	S	S	S	S	S	S	S	S	S	S
Jikedou 2	S	S	S	S	S	S	S	S	S	S	S	S	S	S
Zhonghuang 47	S	S	S	S	S	S	S	S	S	S	S	S	S	S

^1^ R: resistant; S: susceptible.

**Table 2 ijms-20-01809-t002:** Genetic segregation in response to *Phytophthora sojae* isolates PsMC1 and PsJS2 in 137 F_2:3_ families derived from a cross between soybean cultivars Zhonghuang47 and Xiu94-11.

Parent and the Cross	Generation	Total Plants	Observed Number			Except Ratio and Goodness of Fit		
			R ^1^	Rs	S	R:Rs:S	*χ^2^*	*P*
Xiu94-11	P1	15	15	-	-			
Zhonghuang47	P2	15	-	-	15			
Zhonghuang47 × Xiu94-11	F_2:3_	137	38	63	36	1:2:1	0.93	0.62

^1^ R: homozygous resistant; Rs: segregating; S: homozygous susceptible.

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
