# Peer review of "Genetic Mapping and Molecular Characterization of a Broad-spectrum Phytophthora sojae Resistance Gene in Chinese Soybean"

_ijms, 2019, doi:10.3390/ijms20081809_

Reviewer 1 Report

Overall the MS is interesting and informative. I have some concerns which authors can address.

Abstract:

Line 25: "two additional LRR-encoding fragments in the candidate gene.." - is this copy number variation?

Line 27 "may be useful for identifying the RpsX and RpsQ alleles" - not clear why authors are not sure for the markers. "May be" make it doubtful 

From the abstract, it is not clear whether the gene provides a broad spectrum of resistance since the title of the article has highlighted it.

"This study implies that the number of LRR units in the LRR domain may be important for PRR resistance in soybean." - For this, authors expected to provide more experimental evidence if not in the wet lab then at least with computational approaches. 

Introduction 

Line 47 – “partial resistance, which is controlled by multiple genes” – But in the present MS authors are highlighting single gene providing the partial resistance

“More than 30 Rps genes have been identified” – I guess 32 Rps genes have been reported.

Line 57 – “Soybean cultivars containing Rps genes are usually resistant to P. sojae for only 8–15 years. Therefore, researchers must continuously search for and identify new Rps genes.” – Need to rephrase, use the keyword – durability, breakdown of resistance.

Results  

Authors need to improve all the subheadings. Present subheadings in the results sections are more suitable for materials and methods for instance “2.1. Phenotypic Evaluation”. The subheading in the results sections should reflect conclusive result”

“22 other cultivars with a single identified Rps gene as well as four PRR-susceptible cultivars” – are these differential genotypes for Rps, Authors need to use appropriate terms.  

The MS has used four different isolates with the same name “PsRace” which showed different reactions, therefore, better to name it separately, and also need to verify whether genetically those are same or different.

Table 1 need information of Avr genes profile for the P. sojae isolates tested as like Rps profile provided for the soybean genotypes.

Table 3 – “Amount” – change it to “Total plants/genotypes”

Not clear why the authors named it as RpsX – need to follow the world-wide accepted convention for the Rps gene naming.

Figure 1 “The red region represents the identified candidate interval for RpsX.” – the legend does not match to the figure. The black line crosses the threshold several instances other than the RpsX locus.

Author Response

For reviewer1:

Comments and Suggestions for Authors

Overall the MS is interesting and informative. I have some concerns which authors can address.

Abstract:

Line 25: "two additional LRR-encoding fragments in the candidate gene.." - is this copy number variation?

Response: Thanks for your comments. The number of LRR units in the LRR domain plays an important role in resistance to plant disease. Through sequence alignment analysis, we found that the inserted 144-bp fragment has the highest similarity to its adjacent upstream 144-bp sequence, possibly due to unequal exchange. But the sequences of the two 144-bp fragments are not exactly identical, and the 144 bp sequence is not two perfectly intact LRR units, but consists of one complete LRR unit and two incomplete LRR units, which are exactly the same length as the two LRR units, resulting in two more LRR units increasing in the LRR domain region of the gene.

Copy number variation (CNV) is caused by rearrangement of the genome. It generally refers to the increase or decrease of the copy number of large fragments of genomics longer than 1 kb, which mainly due to the loss and duplication of submicroscopic levels (Żmieńko et al. 2014; Saxena et al. 2014). So this small insertion we think may not be copy number variation.

Żmieńko, A., Samelak, A., Kozłowski, P., & Figlerowicz, M. (2014). Copy number polymorphism in plant genomes. Theoretical and applied genetics, 127(1), 1-18.

Saxena, R. K., Edwards, D., & Varshney, R. K. (2014). Structural variations in plant genomes. Briefings in functional genomics, 13(4), 296-307.

Line 27 "may be useful for identifying the RpsX and RpsQ alleles" - not clear why authors are not sure for the markers. "May be" make it doubtful.

Response: Thanks for your detail comments. We have corrected as you suggested in the text. "May be" has been corrected as “proved to be”.

From the abstract, it is not clear whether the gene provides a broad spectrum of resistance since the title of the article has highlighted it.

Response: Thanks for your comments. Broad-spectrum resistance was mentioned in the abstract, but it does not explain how broad-spectrum resistance is expressed. We have corrected the corresponding sentence in the text. Xiu94-11 has broad-spectrum resistance because it is resistant to 14 isolates of P. sojae which conferring the more broad-spectrum resistance than other resistant cultivars in this study.

"This study implies that the number of LRR units in the LRR domain may be important for PRR resistance in soybean." - For this, authors expected to provide more experimental evidence if not in the wet lab then at least with computational approaches.

Response: Thanks for your comments and suggestions. High sequence identity between the sequences encoding the LRR units in the LRR domain, unequal crossing-over and illegitimate recombination are prone to occur, resulting in new R gene specificities due to the differences in the number of encoded LRR units. This has been proved by the previous studies (Lawrence et al. 2010; Thakur et al. 2015). Such as in rice blast resistance gene Pi54, extensive sequence variation in LRR domain makes it confers broad spectrum resistance to rice blast. Similarly with our study, 144-bp consensus region plays an important role in its broad spectrum resistance (Thakur et al. 2015). And also, in resistance genes at the M locus in flax, the difference in the number of LRR units results the emergence of different rust-resistant alleles (Lawrence et al. 2010). Therefore, in this study, the 144-bp insertion of the RpsX candidate genes was caused by an unequal exchange with the adjacent 144-bp fragment, resulting in a new Rps gene.

Thakur S, Singh P K, Das A, et al. Extensive sequence variation in rice blast resistance gene Pi54 makes it broad spectrum in nature[J]. Frontiers in plant science, 2015, 6: 345

Lawrence, GJ., Anderson, PA., Dodds, PN., Ellis, JG. Relationships between rust resistance genes at the M locus in flax. Molecular plant pathology, 2010, 11, 19-32.

Introduction

Line 47 – “partial resistance, which is controlled by multiple genes” – But in the present MS authors are highlighting single gene providing the partial resistance

Response: Thanks for your professional comments. We revised the related sentence for clearly express the meaning in the text.

“More than 30 Rps genes have been identified” – I guess 32 Rps genes have been reported.

Response: Thanks for your comments and suggestions. The corresponding sentence in the text has been corrected.

Line 57 – “Soybean cultivars containing Rps genes are usually resistant to P. sojae for only 8–15 years. Therefore, researchers must continuously search for and identify new Rps genes.” – Need to rephrase, use the keyword – durability, breakdown of resistance.

Response: Thanks for your comments and suggestions. This sentence was corrected as “Due to the breakdown of resistance caused by emergence of new P. sojae pathotypes, durability of an Rps gene is generally only 8–15 years. Therefore, researchers must continuously search for and identify new Rps genes”.

Results 

Authors need to improve all the subheadings. Present subheadings in the results sections are more suitable for materials and methods for instance “2.1. Phenotypic Evaluation”. The subheading in the results sections should reflect conclusive result”

Response: Thanks for your comments and suggestions. We have corrected the subheadings in the text as suggested.

“22 other cultivars with a single identified Rps gene as well as four PRR-susceptible cultivars” – are these differential genotypes for Rps, Authors need to use appropriate terms. 

Response: Thanks for your comments and suggestions. We have corrected the sentence as “22 other cultivars each containing a different identified Rps gene as well as four PRR-susceptible cultivars”.

The MS has used four different isolates with the same name “PsRace” which showed different reactions, therefore, better to name it separately, and also need to verify whether genetically those are same or different.

Response: Thanks for your comments and suggestions. I am sorry for this, because the border of the table covered the names of the different isolates, we have adjusted them.

Table 1 need information of Avr genes profile for the P. sojae isolates tested as like Rps profile provided for the soybean genotypes.

Response: Thanks for your suggestions. PsRace1, PsRace3, PsRace4, PsRace5, Ps41-1 and PsNKI were isolated from Heilongjiang Province, Northeast of China; PsAH4 and PsMC1 were isolated from Anhui province, Huang-Huai Region of China; PsFJ2 and PsFJ3 were isolated from Fujian province in China; PsJS2 was isolated from Jiangsu province of China; PsUSAR2 was the strain of race2 from the United States with pathotype changed (Zhang et al., 2014). Ps6497 and Ps7063 were the standard strains from United States (Dou et al. 2010; Song et al. 2013). To date, most of the isolates of Phytophthora sojae were distinguished on the pathogenic types, and no in-depth study of the Avr gene was performed. Only Ps6494 is reported to contain Avr1k, Avr4 and Avr6 genes (Dou et al. 2010; Song et al. 2013).

Zhang J, Sun S, Wang G, Duan C, Wang X, Wu X, Zhu Z (2014) Characterization of Phytophthora resistance in soybean cultivars/lines bred in Henan province. Euphytica 196: 375-384

Dou D, Kale S D, Liu T, et al. (2010) Different domains of Phytophthora sojae effector Avr4/6 are recognized by soybean resistance genes Rps4 and Rps6[J]. Mol Plant-microbe In 23: 425-435

Song T, Kale S D, Arredondo F D, et al. (2013) Two RxLR avirulence genes in Phytophthora sojae determine soybean Rps1k-mediated disease resistance[J]. Mol Plant-microbe In 26: 711-720

Table 3 – “Amount” – change it to “Total plants/genotypes”

Response: Thanks for your suggestions. The corresponding place in the table has been corrected.

Not clear why the authors named it as RpsX – need to follow the world-wide accepted convention for the Rps gene naming.

Response: Thanks for your suggestions. At present, 12 Rps loci from Rps1 to Rps12 have been officially named internationally. However, most of the Rps genes currently identified in recent years are not based on this naming standard, such as RpsWY, RpsHN, RpsHC18, RpsYD29, RpsJS etc. These newly identified Rps genes are named according to their corresponding cultivars. And the other reason is that many of the Rps genes on chromosome 3 finely mapped are based on the physical position of the soybean reference genome sequence Williams82. Whether these genes are alleles or linked with each other requires further fine mapping, gene cloning or allelic tests. Therefore, for the sake of caution, we named the Rps gene as RpsX identified in soybean cultivar Xiu94-11.

Figure 1 “The red region represents the identified candidate interval for RpsX.” – the legend does not match to the figure. The black line crosses the threshold several instances other than the RpsX locus.

Response: Thanks for your suggestions. The distribution of the delta SNP index at the 99% confidence level revealed only one contiguous region exceeding the threshold in the 1.05–3.55 Mb genomic region (red) of chromosome 3 (Figure 1). The black line crosses the threshold several instances other than the RpsX locus, however, only the value of the delta-SNP-Index (black) of the red region higher than the threshold of 99% confidence level. We have corrected the caption of Figure 1.

Reviewer 2 Report

In this study, the authors make use of QTL-sequencing combined with genetic mapping to identify RpsX  gene, namely Glyma.03g0272000, acting as a broad-spectrum  PRR resistance gene in soybean. Sequencing revealed that 144-bp insertion in  Glyma.03g0272000 resulted in two additional LRR fragment in coding protein.  These results are very interesting, nice, and of importance.  But I still have some concerns.

1. I am curious about the transcript level changes of Glyma.03g0272000 in soybean PRR resistant cultivar(s) when inoculated with Phytophthora sojae at different time point. Moreover it would be better if the authors provided data about the transcript level between two soybean genotypes with different  Phytophthora sojae  tolerance.

2. it would be better if the title is Genetic Mapping and Molecular Characterization of a Broad-spectrum Phytophthora sojae Resistance Gene Glyma.03g027200 in Chinese Soybean.

3. Please correct "a landrace from Taiwan" into  "a landrace from TaiwanProvince' In line 184.

4.  Some typos can be found in the manuscript, for example, in line 223 "Glyma.03g0272000", 

line 274 "Glyma.03g27200", in line 274, 284, and 289 "Glyma.03g27200";  line 373 "Glyma.03g0272000".  These typos should be corrected. Please check the whole manuscript.

Author Response

For reviewer2:

In this study, the authors make use of QTL-sequencing combined with genetic mapping to identify RpsX gene, namely Glyma.03g0272000, acting as a broad-spectrum PRR resistance gene in soybean. Sequencing revealed that 144-bp insertion in Glyma.03g0272000 resulted in two additional LRR fragment in coding protein.  These results are very interesting, nice, and of importance.  But I still have some concerns.

1. I am curious about the transcript level changes of Glyma.03g0272000 in soybean PRR resistant cultivar(s) when inoculated with Phytophthora sojae at different time point. Moreover it would be better if the authors provided data about the transcript level between two soybean genotypes with different Phytophthora sojae tolerance.

Response: Thanks for your comments and suggestions. Analysis of the transcript level between two soybean genotypes is an important part of studying this gene. But I am very sorry that we have not been able to complete the entire analysis of expression analysis in a short period for major review. We will conduct a more in-depth study of the expression and function of this gene in the future.

2. it would be better if the title is Genetic Mapping and Molecular Characterization of a Broad-spectrum Phytophthora sojae Resistance Gene Glyma.03g027200 in Chinese Soybean.

Response: Thanks for your comments and suggestions. However, the name Glyma.03g027200 is an annotated name of gene model on the Williams82 reference genome and does not represent the corresponding allele of Xiu94-11. The allele of Glyma.03g027200 in Xiu94-11 will be further studied and officially named. In addition, the possibility that other types of genes that contribute to broad-spectrum resistance exist within the localization interval of Xiu94-11 cannot be ruled out. Therefore, we temporarily think that the title should not be added to the name Glyma.03g027200.

3. Please correct "a landrace from Taiwan" into "a landrace from Taiwan Province' In line 184.

Response: Thanks for your suggestions. We have corrected it as suggested.

4.  Some typos can be found in the manuscript, for example, in line 223 "Glyma.03g0272000",

line 274 "Glyma.03g27200", in line 274, 284, and 289 "Glyma.03g27200";  line 373 "Glyma.03g0272000".  These typos should be corrected. Please check the whole manuscript.

Response: Thanks for your kind and detailed suggestions. We have corrected corresponding lines in the text and checked typos for the whole manuscript.

Reviewer 3 Report

The manuscript by Zhong et al., reports the identification of a gene that might confer resistance against P. sojae. The identification of this gene was through a novel and powerful technique that combines traditional mapping approach and NGS approaches. The data contained in this study will be well received by the soybean community. Additionally, this manuscript provides valuable information for future soybean breeding.

Although the data is well described and discussed, there are some questions that authors must take into consideration:

1) How is the expression Glyma.03g027200 in the susceptible and resistant line?

2) is it possible to revert the susceptible phenotype by cloning the gene from the resistant parent into the susceptible parent?

Author Response

For reviewer3:

The manuscript by Zhong et al., reports the identification of a gene that might confer resistance against P. sojae. The identification of this gene was through a novel and powerful technique that combines traditional mapping approach and NGS approaches. The data contained in this study will be well received by the soybean community. Additionally, this manuscript provides valuable information for future soybean breeding.

Although the data is well described and discussed, there are some questions that authors must take into consideration:

1) How is the expression Glyma.03g027200 in the susceptible and resistant line?

Response: Thanks for your kind suggestions. Expression analysis of candidate genes is a critical part of the in-depth study of RpsX. However, we were unable to conduct this part of the experiment and analysis within a short period. Next, we will conduct expression analysis of Xiu94-11 and Zhonghuang47 based on your suggestions.

2) is it possible to revert the susceptible phenotype by cloning the gene from the resistant parent into the susceptible parent?

Response: Thanks for your comments and suggestions. Using transgenic methods to transform RpsX into susceptible parental cultivar Zhonghuang47 is the most critical part of the functional study of this gene. The RpsX gene function will be verified and elucidated by the transgenic method and the CRISPR/Cas9-based RpsX functional gene knockout technique in our future study.

Round  2

Reviewer 1 Report

Authors have addressed all of my concerns and improved the MS as expected 

Author Response

Thanks for your comments and suggestions.